# SANER: Annotation-free Societal Attribute Neutralizer for Debiasing CLIP

**Yusuke Hirota**[1,2]*, **Min-Hung Chen**[1], **Chien-Yi Wang**[1], **Yuta Nakashima**[2],
**Yu-Chiang Frank Wang**[1,3], **Ryo Hachiuma**[1]
[1]NVIDIA   [2]Osaka University   [3]National Taiwan University
{y-hirota,nakashima}@is.ids.osaka-u.ac.jp
{minhungc,chienyiw,frankwang,rhachiuma}@nvidia.com

## Abstract

Large-scale vision-language models, such as CLIP, are known to contain societal bias regarding protected attributes (*e.g.,* gender, age). This paper aims to address the problems of societal bias in CLIP. Although previous studies have proposed to debias societal bias through adversarial learning or test-time projecting, our comprehensive study of these works identifies two critical limitations: 1) *loss of attribute information* when it is explicitly disclosed in the input and 2) *use of the attribute annotations* during debiasing process. To mitigate societal bias in CLIP and overcome these limitations simultaneously, we introduce a simple-yet-effective debiasing method called **SANER** (societal attribute neutralizer) that eliminates attribute information from CLIP text features only of *attribute-neutral* descriptions. Experimental results show that SANER, which does not require attribute annotations and preserves original information for *attribute-specific* descriptions, demonstrates superior debiasing ability than the existing methods.[1]

## 1 Introduction

Large-scale vision-language models (VLMs), such as CLIP (Radford et al., 2021), have demonstrated a remarkable capability in multi-modal understanding (Lüddecke & Ecker, 2022; Tewel et al., 2022) and generation (Rombach et al., 2022; Tao et al., 2023; Yamazaki et al., 2023), being trained with million-scale image-text pairs. Utilizing these VLMs, recent vision models have achieved significant performance enhancements across a wide range of computer vision tasks (*e.g.,* captioning (Mokady et al., 2021; Yamazaki et al., 2022; Li et al., 2023b) and object detection (Li et al., 2022; Zhong et al., 2022)), without the necessity for task-specific training (Shen et al., 2022).

Despite the success, several works have identified societal bias regarding demographic attributes, such as gender and age, in these VLMs (Wolfe & Caliskan, 2022; Hausladen et al., 2024; Alabdulmohsin et al., 2024), potentially causing unfair or prejudicial decisions by models. Hall et al. (2023) conducted audits on performance disparity, particularly with respect to gender, and revealed gender-dependency of the CLIP performance. Qiu et al. (2023) also demonstrated that adopting CLIP for caption evaluation tends to favor gender-stereotypical sentences (*e.g.,* preferring "A woman is cooking" over "A man is cooking" for images depicting men), highlighting the inherent gender bias. These findings underscore the importance of addressing bias in VLMs.

Some studies have proposed mitigating societal bias in VLMs (Berg et al., 2022; Seth et al., 2023; Dehdashtian et al., 2024; Chuang et al., 2023). *Adversarial debiasing* (Berg et al., 2022; Seth et al., 2023; Dehdashtian et al., 2024) fine-tunes CLIP to lessen leakage of protected attributes[2] into the features, while *projection-based debiasing* (Chuang et al., 2023) removes the protected attribute encoded in CLIP features at the inference phase. Our holistic review of these pioneering works (Sec. 2), though, identifies the following potential drawbacks or controversies in their design choices.

---

*Work done as an intern at NVIDIA.

[1]Project page: https://rebnej.github.io/saner-clip.github.io/

[2]We refer to any demographic variables, like age and gender, as *protected attribute* (or *attribute* in short), based on which a model's decisions should not be made.

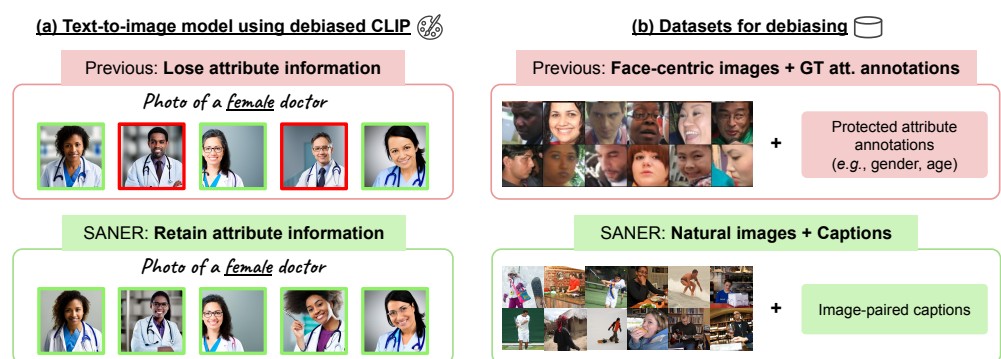

Figure 1: Our debiasing method, SANER, overcomes the limitations in existing methods: (a) attribute information is retained after debiasing, and (b) protected attribute annotations are not required for debiasing.

**Loss of attribute information explicitly disclosed in the input.** Some methods aim to completely remove attribute information by decorrelating the attribute and the features (Dehdashtian et al., 2024) or by squashing the subspace associated with the attribute (Chuang et al., 2023), even when the attribute is explicitly disclosed. This choice can limit the generalizability of a VLM's features to a spectrum of downstream tasks (*e.g.,* Stable Diffusion (Rombach et al., 2022) generates male images for text prompt "a female doctor" when encoded with debiased CLIP as shown in Fig. 1 (a) and Sec. 5.2), while it works for attribute-agnostic downstream tasks (Krause et al., 2013).

**Use of the attribute annotations.** Adversarial debiasing methods (Berg et al., 2022; Seth et al., 2023; Dehdashtian et al., 2024) require protected attribute annotations, as provided in Fair-Face (Karkkainen & Joo, 2021), for fine-tuning (Fig. 1 (b)). Datasets with attribute annotations are still scarce, partly because the annotation process needs ethical considerations (Andrews et al., 2023), limiting their applicability. This dataset scarcity also causes the limited diversity of images and text descriptions used for fine-tuning the VLM, potentially inducing *overfitting*.

This paper presents a simple-yet-effective debiasing approach for CLIP, called **SANER** (societal attribute neutralizer), that simultaneously overcomes the aforementioned limitations. Specifically, SANER trains a debiasing layer (*i.e.,* a multilayer perception) to amend CLIP text feature vectors of *attribute-neutral* descriptions, given by **attribute neutralization**, such that they are equidistant to those of *attribute-specific* descriptions using **annotation-free debiasing loss**. With this, only feature vectors for attribute-neutral descriptions are debiased, whereas the attribute-specific ones retain the original information. Attribute-specific descriptions for all possible attribute groups[3] can be easily augmented by modifying the attribute-specific words in the original descriptions, directing the training without attribute annotations.

**Contribution.** Thanks to our annotation-free debiasing pipeline, SANER is designed to be compatible with any dataset of image-text pairs, such as COCO (Lin et al., 2014). This provides denser guidance for training the debiasing layer compared to the existing methods. Moreover, SANER does not require retraining the CLIP model itself, accessing its original training data, or retraining downstream tasks (*e.g.,* text-to-image generation) when applying the debiased CLIP. Experiments on both discriminative and generative tasks (*i.e.,* text-to-image retrieval (Geyik et al., 2019) and text-to-image generation (Rombach et al., 2022)) show that SANER can mitigate gender, age, and racial biases of CLIP. Moreover, we demonstrate that SANER outperforms the existing methods (Berg et al., 2022; Chuang et al., 2023), showing that SANER leads to less attribute-dependency of the downstream performance while overcoming the limitations in existing methods.

## 2 Review: Existing Debiasing Methods

Several debiasing approaches for CLIP have been introduced, broadly categorized into two main types: *adversarial debiasing* (Berg et al., 2022; Seth et al., 2023; Dehdashtian et al., 2024) and

---

[3]*Attribute group* is a class in a protected attribute (*e.g., female* and *male* in gender).

*projection-based debiasing* (Chuang et al., 2023). This section conducts an in-depth analysis of these existing debiasing strategies, highlighting their respective limitations.

**Notation.** Let $\mathcal{D}$ denote a dataset, each of whose sample is quadruple $(v, t, a, d)$, where $v$ is an image, $t$ is a text description, $a \in \mathcal{A}$ is a protected attribute annotation from set $\mathcal{A}$ of all attribute groups, and $d$ is the ground-truth annotation for a downstream task (if any). The CLIP text and image encoders, denoted by $f_{\mathrm{t}}(t) \in \mathbb{R}^K$ and $f_{\mathrm{v}}(v) \in \mathbb{R}^K$, respectively, take $t$ and $v$ as input and generate corresponding feature vectors in a common space.

## 2.1 ADVERSARIAL DEBIASING

Adversarial debiasing (Berg et al., 2022; Seth et al., 2023; Dehdashtian et al., 2024) aims to eliminate protected attribute information in the CLIP features. Specifically, an adversarial classifier is employed to predict and remove protected attribute $a$ from CLIP features.

**Prompt tuning-based debiasing** (Berg et al., 2022) proposes to use learnable tokens to reduce attribute leakage through the similarity between an image and a set of pre-defined textual concepts. Concretely, for a set $\mathcal{C}$ of pre-defined concepts (*i.e.,* phrases) that are supposed to be attribute non-specific (*e.g., smart* and *attractive*), a sequence $l$ of $k$ learnable tokens are prepended to the sentence template $t_c$ with concept $c \in \mathcal{C}$ (*e.g.,* $t_c =$ "A photo of a smart person" for $c = $ *smart*) to obtain $t'_c = [l, t_c]$, where $[\cdot, \cdot]$ represents sequence concatenation. Then, $t'_c$ and arbitrary $v \in \mathcal{D}$ is fed into the CLIP encoders to compute similarity $s_c(v)$ by

$$s_c(v) = f_{\mathrm{v}}(v)^\top f_{\mathrm{t}}(t'_c). \tag{1}$$

Let $s(v) \in \mathbb{R}^{|\mathcal{C}|}$ denote a vector, each of whose elements is the similarity score $s_c(v)$ for a concept in $\mathcal{C}$. Due to the attribute non-specificity of concepts in $\mathcal{C}$, $s(v)$ should not correlate with attribute $a$ of $v$. However, the CLIP text encoder can embed $a$ into $s(v)$ due to bias, allowing an attribute classifier to predict $a$. We denote the probability of being $a$ given $s(v)$ (or a prediction score of the attribute classifier) by $m_a(s(v))$. Prompt tuning-based debiasing (Berg et al., 2022) uses $m_a$ for adversarial loss, given by

$$\mathcal{L}_{\mathrm{adv}} = -\sum_{v \in \mathcal{D}} \log m_a(s(v)). \tag{2}$$

Minimizing $\mathcal{L}_{\mathrm{adv}}$ with respect to $l$ reduces attribute leakage through $s(v)$. A contrastive loss between image and text features is also used for regularization.

The experiments (Berg et al., 2022) showed that this method could effectively reduce attribute leakage through $f_{\mathrm{t}}(t)$, but the limited number of concepts[4] may limit downstream tasks that enjoy the debiased features because as $l$ is learned only through a sparse set of concepts. Additionally, attribute annotations are necessary for the adversarial loss, resulting in the exclusive use of face-centric image datasets (*e.g.,* FairFace (Karkkainen & Joo, 2021)) as $\mathcal{D}$.

**Additive Residual Learner (ARL)** (Seth et al., 2023) is designed to remove attribute information from CLIP image features. This method assumes that a debiasing layer[5] $r$ can identify a vector to additively amend attribute-neutral image feature vector $\delta(v)$, *i.e.,*

$$\delta(v) = f_{\mathrm{v}}(v) - r(f_{\mathrm{v}}(v)). \tag{3}$$

Similarly to (Berg et al., 2022), an adversarial classifier is trained to predict $a$ from $\delta$ with adversarial loss

$$\mathcal{L}_{\mathrm{adv}} = -\sum_{v \in \mathcal{D}} \log m_a(\delta(v)). \tag{4}$$

The reconstruction loss between $f_{\mathrm{v}}(v)$ and $\delta(v)$ regularizes training to preserve the original features.

This method shares a common limitation with prompt tuning-based debiasing (Berg et al., 2022), notably requiring attribute annotations. Another limitation is that it tries to remove attribute features even when attributes of people in images are explicitly disclosed (*i.e.,* when the person is depicted in an image). Consequently, debiased CLIP is ignorant of protected attributes.

---

[4]Their experiments used 10 concepts.

[5]A fully-connected layer is used.

**Mapper** (Dehdashtian et al., 2024) aims to reduce spurious correlations between attributes $a$ and task label $d$ in $\mathcal{D}$. It applies mappings $f'_\mathrm{v}$ and $f'_\mathrm{t}$ to image and text features, respectively, as $x_\mathrm{v}(v) = f'_\mathrm{v}(f_\mathrm{v}(v))$ and $x_\mathrm{t}(t) = f'_\mathrm{t}(f_\mathrm{t}(t))$ for mitigating dependence on $a$. The adversarial loss is computed using a dependence measure $\mathrm{Dep}(\cdot,\cdot)$ to quantify statistical dependence between features as:

$$\mathcal{L}_\mathrm{adv} = -\mathrm{Dep}(x_\mathrm{v}(v), a) - \mathrm{Dep}(x_\mathrm{t}(t), a). \tag{5}$$

These mapping functions are also trained to maximize the statistical dependence between the features after the mapping and task label $d$, *i.e.,*

$$\mathrm{Dep}(x_\mathrm{v}(v), d) + \mathrm{Dep}(x_\mathrm{t}(t), d), \tag{6}$$

to retain the predictive power on the downstream task while reducing bias.

Similar to (Berg et al., 2022; Seth et al., 2023), Mapper relies on attribute annotations. Moreover, it is designed only to address the spurious correlations between the attribute and task labels for a specific task but not for different tasks.

## 2.2 PROJECTION-BASED DEBIASING

Projection-based debiasing (Chuang et al., 2023) projects CLIP text feature vectors into the orthogonal complement of the space spanned by a set of CLIP text feature vectors that pertain to the protected attribute. Specifically, let $\mathcal{U}$ denote a set of text descriptions with the target attribute (*e.g.,* "A photo of a $w$" $\in \mathcal{U}$, where $w \in \{$"woman", "man"$\}$ for binary gender), and $U$ be a matrix each of whose column vectors is $f_\mathrm{t}(u)$ with $u \in \mathcal{U}$. The projection matrix $P$ into the orthogonal complement for $U$ is given by

$$P = I - U(U^\top U)^{-1} U^\top \tag{7}$$

where $I$ is the identity matrix. $P$ can project a CLIP text feature vector $f_\mathrm{t}(t)$ for a text description $t$ into the orthogonal complement by $P f_\mathrm{t}(t)$. This process removes attribute information by projecting features into the space orthogonal to attribute-specific directions.

Unlike adversarial debiasing, which requires training by gradient descent update, $P$ has a closed-form solution and is computed in the inference phase. However, as with ARL, this method also eliminates attribute information even from descriptions with explicit attributes (*e.g.,* "A photo of a female doctor").

## 2.3 SUMMARY OF THE CHALLENGES

The existing debiasing methods, including adversarial and projection-based, reveal several challenges: 1) **Loss of attribute information** (ARL and projection-based) even with explicit attribute description narrows down the utility of the debiased CLIP. For example, a text-to-image generative model with gender-debiased CLIP features may not properly depict explicitly specified gender (as shown in Sec. 5.2), 2) **Dependency on attribute annotations** (prompt tuning, ARL, and Mapper) constrains the range of datasets that can be utilized, often necessitating the use of face-centric image datasets (*e.g.,* FairFace), as opposed to more diverse, natural image datasets (*e.g.,* COCO).

## 3 SOCIETAL ATTRIBUTE NEUTRALIZER (SANER)

Our method for debiasing CLIP features, SANER, addresses the limitations of the existing methods identified in Section 2. Notably, SANER 1) retains attribute information in cases where the person's attributes are explicitly described and 2) eliminates the reliance on attribute annotations, allowing the use of any image-text dataset for training the debiasing layer.

SANER comprises 1) **attribute neutralization**, which eliminates protected attribute information from input text (Section 3.1); 2) **feature modification**, which removes attribute information from the CLIP text features by amending them with a debiasing layer (Section 3.2); 3) **attribute annotation-free debiasing loss**, ensuring the features are not biased towards any attribute group $g \in \mathcal{A}$ (Section 3.3); and 4) **regularization losses**, which preserve the original CLIP features and the alignment between image and text features (Section 3.4).

Figure 2 shows an overview of SANER. We train the debiasing layer for feature modification over an arbitrary dataset $\mathcal{D} = \{(v, t)\}$ of image $v$ and text description $t$ (*e.g.,* image caption, alt text) pairs, which does not provide attribute annotation $a$ as well as target task label $d$.

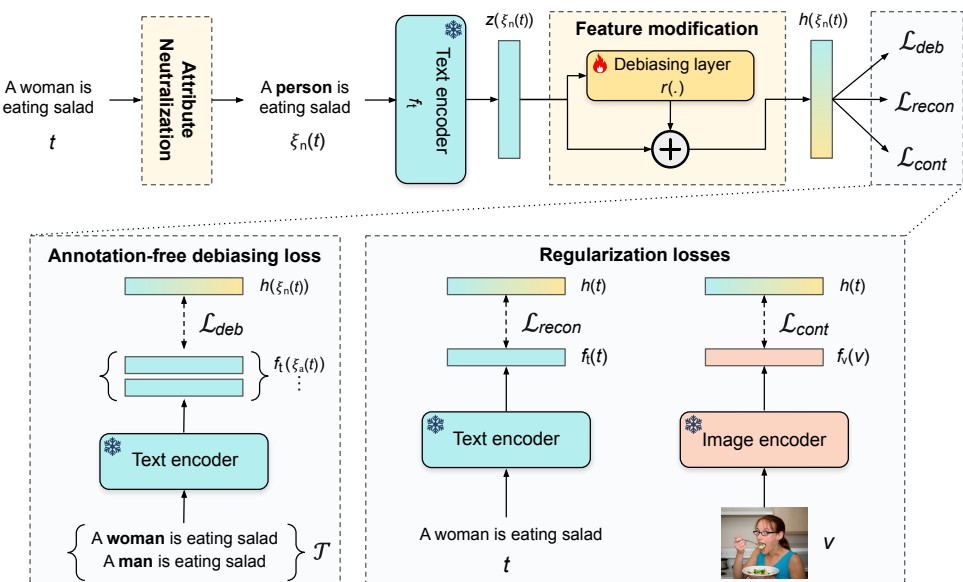

Figure 2: An overview of SANER, exemplified by binary gender. SANER neutralizes attribute-specific text (*e.g.,* "woman" → "person"), modifies features via debiasing layer, and uses three losses for debiasing: $\mathcal{L}_{\text{deb}}$ for attribute neutralization, $\mathcal{L}_{\text{recon}}$ for feature preservation, and $\mathcal{L}_{\text{cont}}$ for image-text alignment.

## 3.1 ATTRIBUTE NEUTRALIZATION

We first modify text description $t \in \mathcal{D}$ that contain person-related words,[6] to remove attribute-specific words. Taking binary gender as a protected attribute,[7] *i.e.,* $\mathcal{A} = \{\texttt{female}, \texttt{male}\}$, as example, the text description

$$t = \text{``A woman is eating salad.''}$$

contains attribute information (*i.e., woman*). We replace attribute-specific terms[8] with the attribute-neutral ones to obtain an attribute-neutral text:

$$\xi_{\text{n}}(t) = \text{``A \underline{person} is eating salad.''}$$

where $\xi_{\text{n}}$ denotes a function for attribute neutralization. Neutralization can be done for other attributes, such as age.[9] We remove age-specific terms[10] (*e.g., young* and *senior*) in text descriptions, for instance, "A young woman is eating salad" → "A woman is eating salad". In contrast to the previous approach (Chuang et al., 2023), which is optimized not to predict the attribute information from the original description $t$, we target the attribute-neutral descriptions $\xi_{\text{n}}(t)$ to preserve the attribute information in the features of attribute-specific descriptions.

## 3.2 FEATURE MODIFICATION

CLIP text features $z(\xi_{\text{n}}(t)) = f_{\text{t}}(\xi_{\text{n}}(t))$ after attribute neutralization can still convey the protected attribute information due to CLIP's bias. To remove such bias, we append a learnable debiasing layer $r$ on top of $f_{\text{t}}$, inspired by recent CLIP fine-tuning techniques (Seth et al., 2023; Gao et al.,

---

[6]Person-related words encompass terms that reference individuals (*e.g., person, girl, man*). The complete list is in the appendix.

[7]Following prior research (Berg et al., 2022; Seth et al., 2023; Dehdashtian et al., 2024; Chuang et al., 2023; Zhao et al., 2017; Garcia et al., 2023; Burns et al., 2018), we focus on the binary gender but recognize the importance of inclusivity. SANER applies to non-binary genders.

[8]We use gender words defined in (Hirota et al., 2023).

[9]Examples for the race attribute are in the appendix.

[10]We define age-specific terms. The list is in the appendix.

2024). Neutralized $t$'s debiased feature $h(\xi_\mathrm{n}(t))$ is given by

$$h(\xi_\mathrm{n}(t)) = z(\xi_\mathrm{n}(t)) + r(z(\xi_\mathrm{n}(t))). \tag{8}$$

### 3.3 ATTRIBUTE ANNOTATION-FREE DEBIASING LOSS

To train $r$ to extract attribute information from CLIP features without attribute annotations, we create a set $\mathcal{T}$ of attribute-specific descriptions for $t \in \mathcal{D}$ and for $g \in \mathcal{A}$, *i.e.*, $\mathcal{T} = \{\xi_g(t)|t \in \mathcal{D}, g \in \mathcal{A}\}$, where $\xi_g(t)$ generates a description for each attribute group $g$ from $t$. For binary gender, this involves generating descriptions with female- and male-specific words. For instance, from the text description, "A woman is eating salad.", we generate two sentences with female and male attributes:



A woman is eating salad.  
A man is eating salad.



The debiasing loss trains $r$ such that $h(\xi_\mathrm{n}(t))$ is equidistant from $f_\mathrm{t}(\xi_g(t))$ for all attribute groups in $\mathcal{A}$, ensuring an impartial representation across the spectrum of attribute groups. We implement this loss as the standard deviation of the cosine similarity between $h(\xi_\mathrm{n}(t))$ and $f_\mathrm{t}(\xi_g(t))$. Let $s_g(t)$ denote the similarity, *i.e.,*

$$s_g(t) = \frac{h(\xi_\mathrm{n}(t))^\top f_\mathrm{t}(\xi_g(t))}{\|h(\xi_\mathrm{n}(t))\| \, \|f_\mathrm{t}(\xi_g(t))\|}. \tag{9}$$

The debiasing loss $\mathcal{L}_\mathrm{deb}$ is defined as

$$\mathcal{L}_\mathrm{deb} = \sqrt{\frac{1}{|\mathcal{D}|} \sum_{t \in \mathcal{D}} (s_g(t) - \bar{s}(t))^2}, \tag{10}$$

where $\bar{s}(t) = \sum_{g \in \mathcal{A}} s_g(t)/|\mathcal{A}|$. A lower standard deviation means $s_g$ is close to $\bar{s}$, leading to $h(\xi_\mathrm{n}(t))$ being equidistant to $f_\mathrm{t}(\xi_g(t))$ for all $g \in \mathcal{A}$. Notably, this debiasing loss can be computed without attribute annotations.

### 3.4 REGULARIZATION LOSSES

Applying the debiasing loss alone significantly changes original CLIP features, thereby losing semantics (Berg et al., 2022; Seth et al., 2023). To maintain the alignment of resulting image-text features, we utilize reconstruction loss (Seth et al., 2023) and contrastive loss (Radford et al., 2021). Reconstruction loss $\mathcal{L}_\mathrm{recon}$ is the mean squared error between $f_\mathrm{t}(t)$ and $h(t)$. Contrastive loss $\mathcal{L}_\mathrm{cont}$ aims to minimize the negative log-likelihood of input image-caption pairs, $f_\mathrm{v}(v)$ and $f_\mathrm{t}(t)$, in comparison to negative ones. Note that the original description $t$ is used for regularization losses.

### 3.5 TRAINING AND INFERENCE

The overall loss $\mathcal{L}$ is given by:

$$\mathcal{L} = \alpha \mathcal{L}_\mathrm{deb} + \beta \mathcal{L}_\mathrm{recon} + \gamma \mathcal{L}_\mathrm{cont}, \tag{11}$$

where $\alpha$, $\beta$, and $\gamma$ are the hyperparameters to weight respective losses.

During inference, we apply the trained debiasing layer $r$ and use the modified text features $r(f_\mathrm{t}(t))$ as the CLIP text features.

## 4 EXPERIMENTS: TEXT-TO-IMAGE RETRIEVAL

Following previous studies (Berg et al., 2022; Seth et al., 2023; Dehdashtian et al., 2024; Chuang et al., 2023), we evaluate SANER on the text-to-image retrieval task regarding gender, age, and racial biases. Further analysis, such as the ablation study of the loss components, is in the appendix.

Table 1: **Gender bias**, evaluated by MaxSkew@1000 (scaled by 100), on FairFace and PATA for the original CLIP (Original), prompt tuning-based debiasing (Prompt), projection-based debiasing (Projection), and our method (SANER). A lower value is better (less gender bias). **Bold** represents the best across the models.

| CLIP Model | FairFace | | | PATA | | |
|---|---|---|---|---|---|---|
| | Adjective | Occupation | Activity | Adjective | Occupation | Activity |
| Original (Radford et al., 2021) | 22.9 | 33.7 | 19.5 | 12.1 | 18.7 | 10.7 |
| Prompt (Berg et al., 2022) | 12.3 | 29.9 | 20.0 | 6.7 | 16.5 | 10.2 |
| Projection (Chuang et al., 2023) | 15.4 | 37.4 | 15.0 | 6.4 | 13.6 | 5.4 |
| SANER (Ours) | **8.9** | **14.5** | **7.7** | **5.4** | **9.5** | **3.3** |

## 4.1 EXPERIMENTAL SETTINGS

**Evaluation metric.** We employ the **MaxSkew** metric (Geyik et al., 2019), utilized in the previous studies (Berg et al., 2022; Seth et al., 2023; Dehdashtian et al., 2024; Chuang et al., 2023), to quantify the societal bias in CLIP in the text-to-image retrieval task. MaxSkew measures the disparity between the attribute distribution of $|\mathcal{A}|$ in the top-$k$ retrieved images. Let $\eta_{ak}(q)$ denote the ratio of images labeled with attribute $a$ in the top-$k$ retrieved images. For attribute neutral query $q$, $\eta_{ak}(q)$ should be $1/|\mathcal{A}|$ if the model is unbiased. MaxSkew@$k$ is defined as:

$$\text{MaxSkew@}k = \max_{a \in \mathcal{A}} \log \frac{\eta_{ak}(q)}{1/|\mathcal{A}|}. \tag{12}$$

Ideally, MaxSkew@$k$ is 0 but is larger when a model biased.

**Evaluation setting.** For the attribute-neutral queries, we use template-based queries such as "a photo of a $c$ person", where $c$ is an attribute-neutral concept. Prior work (Berg et al., 2022) has defined a list of person-related **adjectives**, such as *clever* and *attractive*, as attribute-neutral concepts. We extend the list to encompass **occupations** (*e.g., doctor*, *nurse*), and **activities** (*e.g., cooking* and *cleaning*) for a more comprehensive evaluation. MaxSkew@$k$ is computed per concept.[11]

We also evaluate **zero-shot image classification accuracy** on ImageNet-1K (Russakovsky et al., 2015) to ensure that debiasing does not spoil the original CLIP's performance.

**Evaluation datasets.** We utilize two datasets, FairFace (Karkkainen & Joo, 2021) and PATA (Seth et al., 2023), which consist of images alongside protected attribute annotations (*e.g., female* and *male* for gender) associated with the person in each image. FairFace consists of $10,954$ cropped face-centric images, while PATA contains $4,934$ natural images with a single person. Most debiasing approaches only report the performance on the FairFace dataset, but we additionally employ PATA to evaluate the debias performance on more diverse images.

**Methods for comparison.** We compare SANER against existing methods, *i.e.,* prompt tuning-based debiasing (Berg et al., 2022) and projection-based debiasing (Chuang et al., 2023), whose code is publicly available. Unfortunately, we could not reproduce the other methods(Seth et al., 2023; Dehdashtian et al., 2024) since sufficient reproduction details are unavailable.

**Implementation details.** Following the previous works (Berg et al., 2022; Chuang et al., 2023), we employ CLIP (Radford et al., 2021) with ViT-B/16 backbone (Dosovitskiy et al., 2021) as a target model in our experiments. We train the debiasing layer (a multilayer perceptron with two linear layers with ReLU activation (Nair & Hinton, 2010)) appended to the model using $170,624$ image-caption pairs, which is a subset of the COCO training set (Lin et al., 2014) with person-related words/phrases (*e.g., person* and *boy*). We empirically set $\alpha$, $\beta$, and $\gamma$ to 1.0, 0.1, and 0.0001 (Eq. 11), respectively, and train for 5 epochs. Further details are provided in the appendix.

## 4.2 GENDER BIAS ANALYSIS

Table 1 presents MaxSkew@1000 on FairFace and PATA for gender bias, showing SANER mitigates bias the most among all methods. In contrast to the existing methods, this tendency is consistent across 1) datasets with different image domains (*i.e.,* face-centric and natural images) and 2) concept types (*i.e.,* adjective, occupation, and activity). For instance, while the prompt tuning-based method

---

[11]The complete lists of the concepts are in the appendix.

Table 2: **Age bias** (left) and **racial bias** (right), evaluated by MaxSkew@1000 (scaled by 100), on FairFace. **Bold** denotes the best across the models. Results on PATA are in the appendix.

| CLIP Model | FairFace | | | CLIP Model | FairFace | | |
|---|---|---|---|---|---|---|---|
| | Adjective | Occupation | Activity | | Adjective | Occupation | Activity |
| Original | 111.1 | 121.1 | 113.0 | Original | 62.2 | 57.4 | 68.3 |
| Projection | 107.6 | 112.8 | **100.0** | Projection | 56.9 | 75.3 | 67.0 |
| SANER (Ours) | **96.0** | **112.6** | 101.9 | SANER (Ours) | **49.3** | **45.7** | **46.6** |

(Berg et al., 2022) fails to mitigate bias for activity concepts on FairFace (*i.e.,* bias is amplified from 19.5 to 20.0), SANER significantly mitigates bias from 19.5 to 7.7. This verifies the better debiasing performance of SANER compared to the existing methods on diverse concepts and image domains, possibly because SANER is trained with diverse text descriptions (*i.e.,* captions in COCO), which are not constrained like pre-defined concepts required in the previous method (Berg et al., 2022).

### 4.3 AGE AND RACIAL BIASES ANALYSIS

Tables 2 presents the results of MaxSkew@1000 for age and racial biases. We compare SANER with projection-based debiasing (Chuang et al., 2023), as the prompt tuning-based method (Berg et al., 2022) does not provide age and race debiasing variants. Similarly to the results for gender bias, SANER surpasses the existing method across the datasets and the concept types. For example, SANER successfully mitigates the racial bias on FairFace. Meanwhile, the projection-based method amplifies the bias on the occupation concept (*i.e.,* from 57.4 to 75.3). These results validate the generalizability of SANER in bias mitigation across the protected attributes.

### 4.4 ZERO-SHOT IMAGE CLASSIFICATION

To verify whether debiasing harms the zero-shot image classification performance of the original CLIP, we evaluate the prompt tuning-based method (Berg et al., 2022) and SANER on ImageNet-1K in terms of classification accuracy. Projection-based debiasing (Chuang et al., 2023) does not apply to this evaluation because zero-shot prompts, such as "a photo of a car," do not necessarily include person-related words.[12] The results are

Table 3: Accuracy on Image-Net-1K.

| CLIP Model | Acc. (%) |
|---|---|
| Original | 65.4 |
| Prompt | 64.1 |
| SANER (Ours) | 65.2 |

shown in Tab. 3, showing that applying SANER maintains classification performance, whereas the performance of the prompt tuning-based method slightly degrades.

> **Insight**: Our method, SANER, which does not require attribute annotations, outperforms previous methods—including those that do require annotations—in mitigating gender, age, and racial biases without compromising classification performance.

## 5 EXPERIMENTS: TEXT-TO-IMAGE GENERATION

We also evaluate SANER on text-to-image generation, for which societal bias is actively investigated (Bansal et al., 2022; Cho et al., 2023; Liu et al., 2024b). Specifically, we conduct two experiments from different aspects: 1) gender bias regarding occupations using gender-neutral prompts (Sec. 5.1), and 2) retention of attribute information when prompts explicitly disclose gender (Sec. 5.2).

**Image generation settings.** We use Stable Diffusion (SD) v2.1 (Rombach et al., 2022) as the text-to-image generation model. The CLIP text encoder in SD is replaced with the debiased one for evaluation. Following (Chuang et al., 2023), we use gender-neutral prompts with specifying occupations to analyze gender bias in generated images. These prompts are derived from the template "A photo of a *o*", where *o* is replaced with specific occupation terms, such as *doctor* and *teacher*.[13] On the other hand, we employ gender-specific prompts to evaluate the capability of attribute information retention. Concretely, a gender term, either *female* or *male*, is added just before the occupation

---

[12]Projection-based debiasing requires modified input text to include attribute terms.

[13]The list of the occupations is in the appendix.

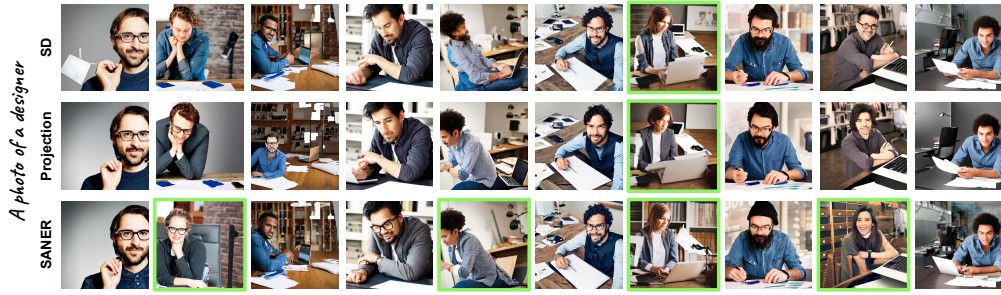

Figure 3: Generated images for the prompt, "A photo of a designer," by the original Stable Diffusion (SD), projection-based debiased CLIP (Projection), and our debiased CLIP (SANER). We randomly sample 10 images from generated images. Images framed in green denote those of the minority gender in the generated images (*i.e.,* female).

terms, *i.e.,* "A photo of a {female/male} $o$", to see if generated images specify the gender. We generate 100 images for each prompt with SD's default hyperparameters.

**Evaluation metrics.** For the bias evaluation for the generative task, we use **statistical parity** (SP) metric that measures the disparity of attribute groups in generated images (Teo et al., 2023; Chuang et al., 2023). Specifically, we annotate binary gender labels (*i.e.,* $\mathcal{A} = \{\texttt{female}, \texttt{male}\}$) for the generated images with the assistance of human workers.[14] SP is defined as the difference between the empirical distribution $\kappa_a$ of gender $a$ and uniform distribution, given by:

$$\text{SP} = \sqrt{\sum_{a \in \mathcal{A}} (\kappa_a - 1/|\mathcal{A}|)^2}, \tag{13}$$

where $\kappa_a = N_a / \sum_{a'} N_{a'}$ with $N_a$ being the number of images annotated as $a$. For an unbiased text-to-image generation model, SP should be 0 but increases for biased models.

For gender-specific prompts for evaluating gender information retention, we compute the **accuracy**, *i.e.,* the ratio of the images that contain the same gender as the prompt to all generated images.

## 5.1 GENDER BIAS ANALYSIS

Table 4 summarizes SP scores, which demonstrate that applying SANER to Stable Diffusion notably mitigates gender bias regarding occupations. SANER again outperforms projection-based debiasing one (*i.e.,* 0.39 for SANER and 0.47 for projection-based), highlighting the superiority of SANER.

We show visual examples where SANER mitigates gender bias in Fig. 3. For the gender-neutral prompt, "A photo of a designer," the original SD and projection-based debiasing (Projection) predominantly generate images of a man. In contrast, ours shows a more balanced gender distribution.

Table 4: Gender bias (SP) and gender information retention (Accuracy) in images from Stable Diffusion (Original) and the model using projection-based debiased CLIP (Projection) and our debiased CLIP (SANER). Results are the mean across occupations. Female and male refer to prompts specifying each gender. **Bold** indicates the best.

| CLIP Model | SP ↓ | Accuracy ↑ Female | Accuracy ↑ Male |
|---|---|---|---|
| Original | 0.51 | **1.00** | **1.00** |
| Projection | 0.47 | 0.58 | 0.79 |
| SANER (Ours) | **0.39** | **1.00** | **1.00** |

## 5.2 ASSESSMENT OF RETENTION OF ATTRIBUTE INFORMATION

Table 4 also shows the accuracy of how much the gender of the person in generated images matches the gender specified in the prompts. For both prompts that describe women and men, using debiased CLIP by projection-based debiasing leads to losing gender information (*i.e.,* accuracies for projection are much lower than those for the original). Conversely, SANER retains gender information (*i.e.,* accuracies for SANER are 1.00). Figure 4 confirms this, showing that using projection-based debiased CLIP results in generating male images for the prompt, "A photo of a female doctor".

---

[14]Different from previous works that use pre-trained gender classifiers to assign gender labels, we do not use them due to their bias issues (Ramaswamy et al., 2021; Das et al., 2018; Dinan et al., 2020).

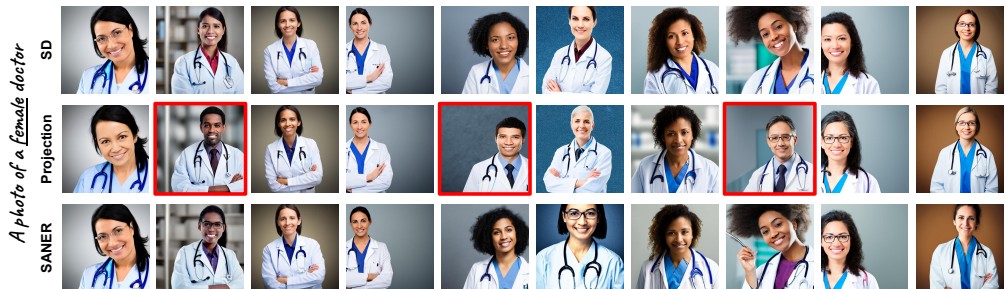

Figure 4: Generated images for the prompt, "A photo of a female doctor," by the original Stable Diffusion (SD), projection-based debiased CLIP (Projection), and our debiased CLIP (SANER). Red frame indicates images with incorrect gender (*i.e.,* male).

> **Insight**: Applying SANER-debiased CLIP to Stable Diffusion generates images with a more gender-equal distribution while preserving gender information in input texts.

## 6 LIMITATIONS

**Further bias mitigation.** Our experiments show that SANER noticeably reduces bias in CLIP. Nonetheless, the bias is not completely *eliminated* (*e.g.,* MaxSkew is not zero). A promising direction for further debiasing could involve debiasing the image encoder, specifically training a debiasing layer to remove attribute information from the visual features for images without human subjects.

**Intersectional bias analysis.** While our experiments focus on gender, age, and racial biases individually, following prior works (Berg et al., 2022; Seth et al., 2023; Dehdashtian et al., 2024; Chuang et al., 2023), SANER can be easily extended to various protected attributes and their combinations. For instance, considering the intersection of binary gender and age, we generate four sentences with (*female*, *young*), (*female*, *old*), (*male*, *young*), and (*male*, *old*) for the debiasing loss, *e.g.,* "A young woman is eating salad" for the input text "A woman is eating salad". This potential for addressing complex biases is noted in future research.

**Use of pre-defined attribute words.** While SANER requires general lists of attributes, creating attribute lists is a one-time effort requiring minimal resources, compared to the ongoing cost and complexity of dataset annotation that often needs ethical review and domain expertise.

## 7 RELATED WORK

As VLMs like CLIP are applied to more tasks, concerns about their social biases have grown (Dehouche, 2021; Wang et al., 2021; Ross et al., 2021; Wolfe et al., 2023; Ruggeri et al., 2023; Srinivasan & Bisk, 2022; Hirota et al., 2024a;b). Hall *et al.* (Hall et al., 2023) examined gender bias in CLIP, uncovering significant discrepancies in object recognition performance based on the gender depicted in images. such as higher accuracy in recognizing an *umbrella* with *women* than with *men*. These biases risk reinforcing discrimination against marginalized groups (Qiu et al., 2023; Tanjim et al., 2024; Wang et al., 2023). Birhane *et al.* (Birhane et al., 2021) demonstrated that adopting CLIP-based filtering in the dataset creation can select stereotypical images, like labeling a female astronaut as a "smiling housewife," leading to harmful, biased datasets.

## 8 CONCLUSION

This paper proposed SANER, a simple-yet-effective debiasing method for CLIP, consisting of **attribute neutralization** and **anotation-free debiasing loss**. Consequently, SANER can leverage any image-text dataset to train the debiasing layer, outperforming existing methods in both discriminative and generative tasks (*i.e.,* text-to-image retrieval and text-to-image generation). We also confirmed that SANER retains attribute information for attribute-specific descriptions through the gender-specified prompts for text-to-image generation.

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

APPENDIX

This appendix includes:

- Implementation details for SANER (Appendix A).
- Further analysis (Appendix B).
- List of person-, gender-, age-, and race-specific terms (Appendix C).
- List of concepts and occupations (Appendix D).
- Additional visual examples for image generation experiments (Appendix E).
- Potential extension (Appendix F).
- Potential negative impact (Appendix G).

## A  IMPLEMENTATION DETAILS FOR SANER

We train the debiasing layer (a multilayer perception with two linear layers with ReLU activation) using $170,624$ image-caption pairs, which is a subset of the COCO training set with person-related words/phrases defined in Sec. C. The hidden embedding dimensionality for the debiasing layer is set to 128. We empirically set $\alpha$, $\beta$, and $\gamma$ to 1.0, 0.1, and 0.0001 (Eq. (11) in the main paper), respectively. We set the training epochs, batch size, and learning rate to 5, 128, and $5 \times 10^{-6}$, respectively. The training is conducted with a machine equipped with a single NVIDIA A100 GPU 40GB, and it took five hours to train the debiasing layer. Note that the weights of the debiasing layer are updated, and the weights for the rest of the modules are frozen.

**Attribute-specific description generation.** To avoid computational complexity, we implement a systematic approach using predefined mapping dictionaries between gender terms (*e.g.,* "woman" $\rightarrow$ "man", "she" $\rightarrow$ "he"). Our algorithm carefully identifies and replaces gender-specific tokens while preserving sentence structure, ensuring no ungrammatical combinations (like "A he/she") are generated. This ensures efficient and coherent text generation that maintains natural language patterns.

**Racial bias mitigation.** For race attribute, we remove race-specific terms[15] (*e.g., African* and *Asian*) in text descriptions, for instance, "An African woman is eating salad" $\rightarrow$ "A woman is eating salad".

## B  ADDITIONAL EXPERIMENTS

### B.1  COMPLETE RESULTS FOR AGE AND RACIAL BIASES

In Table 5 and 6, we show the complete results of Table 2, including results on PATA. The results further verify that SANER demonstrates superior performance in mitigating age and racial biases compared to the existing method.

### B.2  LOSS ABLATION

To validate the effectiveness of each regularization loss (*i.e.,* reconstruction loss and contrastive loss), we conduct an ablation study by removing one of the losses or both losses. Table 7 presents the results of gender bias. The results show that using both reconstruction and contrastive losses yields the best results regarding gender bias mitigation and zero-shot classification accuracy on ImangeNet (Russakovsky et al., 2015). Furthermore, SANER without regularization losses significantly degrades CLIP's zero-shot classification ability (*i.e.,* from 65.4 to 58.8). These observations confirm the importance of having both reconstruction and contrastive losses for the regularization.

**Negative impact on adding only the contrastive loss.** When using only the contrastive loss, MaxSkew scores on FairFace increase compared to no regularization (18.0/19.4/21.7 vs. 15.7/15.0/15.3 without regularization), indicating less effective debiasing. This occurs because the

---

[15]We define race-specific terms. The list is in Section C.

Table 5: **Age bias**, evaluated by MaxSkew@1000 (scaled by 100), on FairFace and PATA. **Bold** denotes the best across the models.

| CLIP Model | FairFace | | | PATA | | |
|---|---|---|---|---|---|---|
| | Adjective | Occupation | Activity | Adjective | Occupation | Activity |
| Original(Radford et al., 2021) | 111.1 | 121.1 | 113.0 | 40.4 | 44.4 | 39.7 |
| Projection (Chuang et al., 2023) | 107.6 | 112.8 | **100.0** | 37.6 | 45.6 | 45.5 |
| SANER (Ours) | **96.0** | **112.6** | 101.9 | **30.3** | **36.7** | **27.5** |

Table 6: **Racial bias**, evaluated by MaxSkew@1000 (scaled by 100), on FairFace and PATA. **Bold** denotes the best across the models.

| CLIP Model | FairFace | | | PATA | | |
|---|---|---|---|---|---|---|
| | Adjective | Occupation | Activity | Adjective | Occupation | Activity |
| Original(Radford et al., 2021) | 62.2 | 57.4 | 68.3 | 33.4 | 28.6 | 31.5 |
| Projection (Chuang et al., 2023) | 56.9 | 75.3 | 67.0 | **19.9** | 43.8 | 26.3 |
| SANER (Ours) | **49.3** | **45.7** | **46.6** | 28.9 | **21.2** | **20.5** |

contrastive loss alone focuses on maintaining image-text alignment but does not constrain the debiased features to remain close to the original CLIP features. As a result, the feature modifications may become suboptimal for bias removal. The reconstruction loss plays a crucial role in ensuring the modified features preserve essential semantic information while effectively removing unwanted bias.

### B.3 ANALYSIS ON THE DATA SIZE

The experiments in the main paper (Sections 5 and 6) verify that SANER outperforms existing debiasing methods, showing a better bias mitigation ability in terms of gender and age biases. This superior performance of SANER may be because SANER is trained with diverse text descriptions (*i.e.,* captions in COCO (Lin et al., 2014)), which are not constrained like pre-defined concepts required in the previous method (Berg et al., 2022). In this section, we conduct an experiment to verify this hypothesis. Specifically, we use $n$ percent of the training samples (*i.e.,* $17,624$ image-caption pairs of COCO) to evaluate the impact of the training dataset size. We use the same settings in Sec. A, but use the different training epochs to align the number of iterations. The results are shown in Table 8.

The results validate that as the number of data samples increases, gender bias is reduced. Specifically, while using a part of the training data results in mitigating gender bias (*i.e.,* MaxSkew scores are smaller than the original CLIP), using the full training samples (*i.e.,* COCO-100%) gives the best results, showing the importance of the use of more diverse data for debiasing.

### B.4 QUALITY OF THE GENERATED IMAGES

We evaluate image fidelity and image-text alignment on the COCO Karpathy test set. Specifically, we use FID score (Heusel et al., 2017) and CLIPScore (Hessel et al., 2021) to measure fidelity and image-text alignment, respectively. The results show that SANER achieves comparable performance to the original Stable Diffusion (FID: 28.1, CLIPScore: 31.2 vs. FID: 28.7, CLIPScore: 31.0 for SANER), confirming that SANER achieves debiasing while maintaining image generation quality.

### B.5 EXPERIMENTS ON BLIP

To verify the effectiveness of SANER for VLMs beyond CLIP, we conduct gender bias experiments using BLIP (Li et al., 2023a). As shown in Table 9, SANER demonstrates superior debiasing performance compared to the original BLIP and the projection-based debiasing.

### B.6 EXPERIMENTS ON FACET

In addition to FairFace and PATA, we evaluate SANER on the FACET dataset (Gustafson et al., 2023). Similar to PATA, FACET comprises real-world, natural images but includes a broader range

Table 7: Gender bias, evaluated by MaxSkew@1000 (scaled by 100), on FairFace and PATA for our method (SANER) with different regularization loss combinations. Recon denotes the use of the reconstruction loss, and cont represents the use of the contrastive loss. IN acc is the zero-shot classification accuracy on ImageNet. Adj, Occ, and Act represent the types of concepts (*i.e.,* Adjective, Occupations, and Activity, respectively). A lower value is better (less gender bias). **Bold** represents the best across the SANER variants.

| CLIP Model | FairFace | | | PATA | | | IN acc (%) |
|---|---|---|---|---|---|---|---|
| | Adj | Occ | Act | Adj | Occ | Act | |
| Original (Radford et al., 2021) | 22.9 | 33.7 | 19.5 | 12.1 | 18.7 | 10.7 | 65.4 |
| SANER | | | | | | | |
| (no regularization) | 15.7 | 15.0 | 15.3 | 15.8 | 14.4 | 15.4 | 58.8 |
| Recon | 10.2 | 16.7 | 11.4 | 6.2 | 12.0 | 6.2 | 64.2 |
| Cont | 18.0 | 19.4 | 21.7 | 7.0 | **9.1** | 9.0 | 63.0 |
| Recon+Cont | **8.9** | **14.5** | **7.7** | **5.4** | 9.5 | **3.3** | **65.2** |

Table 8: Gender bias, evaluated by MaxSkew@1000 (scaled by 100), on FairFace and PATA for the original CLIP (Original) and our method (SANER) with different data sizes. COCO-$n$% denotes that we use $n$% of the training samples. IN acc is the zero-shot classification accuracy on ImageNet. A lower value is better (less gender bias). **Bold** represents the best across the SANER variants.

| CLIP Model | FairFace | | | PATA | | | IN acc (%) |
|---|---|---|---|---|---|---|---|
| | Adj | Occ | Act | Adj | Occ | Act | |
| Original (Radford et al., 2021) | 22.9 | 33.7 | 19.5 | 12.1 | 18.7 | 10.7 | 65.4 |
| SANER | | | | | | | |
| COCO-50% | 15.7 | 19.4 | 17.9 | 10.0 | 14.1 | 10.2 | **65.4** |
| COCO-75% | 12.8 | 17.6 | 16.1 | 7.8 | 13.0 | 8.3 | **65.4** |
| COCO-100% | **8.9** | **14.5** | **7.7** | **5.4** | **9.5** | **3.3** | 65.2 |

of annotations for protected attributes, such as hair color. To evaluate SANER's effectiveness on FACET, which differs in data distribution from FairFace and PATA, we conducted experiments focusing on gender bias. The results, presented in Table 10, demonstrate SANER's superior debiasing performance, further supporting its robustness across diverse dataset distributions. Experiments on other attributes, such as hair color, are left for future work.

## C  LIST OF PERSON-, GENDER-, AGE-, RACE-SPECIFIC TERMS

The **person-related words** that are used to identify text descriptions that are relevant to humans (in Sec. 4.1 in the main paper) are as below:

*actor, actress, adult, architect, artist, associate, aunt, baby, boy, boyfriend, brother, chairman, chairperson, chairwoman, chef, child, coach, colleague, comedian, counselor, cowboy, cowgirl, dancer, daughter, designer, director, doctor, driver, dude, elder, emperor, employee, employer, engineer, entrepreneur, executive, expecting, father, female, friend, gentleman, girl, girlfriend, guy, hairdresser, he, her, hers, herself, him, himself, his, husband, individual, infant, instructor, kid, lady, lawyer, leader, lecturer, male, man, manager, mechanic, member, mentor, mother, musician, neighbor, novelist, nurse, parent, partner, people, performer, person, pharmacist, photographer, physician, pilot, player, police officer, policeman, policewoman, politician, pregnant, prince, princess, professor, queen, relative, researcher, royal, scholar, scientist, secretary, server, she, sibling, singer, sister, son, specialist, spouse, student, surfer, surgeon, tailor, teacher, technician, teenager, their, theirs, them, themselves, therapist, they, toddler, uncle, veterinarian, volunteer, waiter, waitress, wife, woman, worker, writer, youth,* and their plurals.

We list the **gender-specific terms** that are used to create attribute-neutral text descriptions $\xi_n(t)$ (in Sec. 4.1 in the main paper): *woman, female, lady, mother, girl, aunt, wife, actress, princess, waitress, sister, queen, pregnant, daughter, she, her, hers, herself, man, male, father, gentleman, boy, uncle, husband, actor, prince, waiter, son, brother, guy, emperor, dude, cowboy, he, his, him, himself* and their plurals (orange denotes female-specific words, and olive represents male-specific terms). To synthesize attribute-specific descriptions (*i.e.,* $\mathcal{T} = \{\xi_g(t)|t \in \mathcal{D}, g \in \mathcal{A}\}$ in Sec.

Table 9: Gender bias for **BLIP**, evaluated by MaxSkew@1000.

| BLIP Model | FairFace | | | PATA | | |
|---|---|---|---|---|---|---|
| | Adjective | Occupation | Activity | Adjective | Occupation | Activity |
| Original | 16.8 | 15.3 | 12.8 | 7.7 | 7.4 | 5.5 |
| Projection | **14.9** | 19.8 | 17.4 | 12.0 | 14.7 | 17.4 |
| SANER (Ours) | 15.3 | **13.9** | **10.9** | **6.7** | **6.7** | **4.9** |

Table 10: Gender bias on **FACET**, evaluated by MaxSkew@1000.

| CLIP Model | FACET | | |
|---|---|---|---|
| | Adjective | Occupation | Activity |
| Original | 46.0 | 37.7 | 33.4 |
| Projection | 37.3 | 46.3 | 41.1 |
| SANER (Ours) | **31.7** | **24.7** | **25.2** |

4.3 in the main paper) for binary gender, we replace person-specific terms in the attribute-neutral descriptions with their corresponding gender terms (*e.g., person → woman* and *person → man*).

We also list the **age-specific terms** used to create attribute-neutral text descriptions $\xi_n(t)$ (in Sec. 4.1 in the main paper): *elderly ,baby*, *child*, *kid*, *teenager*, *adult*, *youth*, *infant*, *toddler*, *elder*, *girl*, *boy*, *young*, *old*, *teenage*, and their plurals. To create attribute-specific descriptions (*i.e.,* $\mathcal{T} = \{\xi_g(t)|t \in \mathcal{D}, g \in \mathcal{A}\}$ in Sec. 4.3 in the main paper) for binary age, we add *young* or *old* just before the person-specific terms (*e.g., person → young person* and *person → old person*).

Regarding race attributes, we use **race-specific** terms to create attribute-neutral text descriptions: *african*, *africa*, *asian*, *oriental*, *asia*, *east asian*, *south asian*, *south east asian*, *black*, *caucasian*, *european*, *hispanic*, *latino*, *latina*, *latinx*, *white*, *arab*, *arabic*, *middle eastern*, *native*, *indigenous*, *american*, *african american*, *usa*, *united states*, *chinese*, *china*, *japanese*, *japan*, *indian*, *india*, *mexican*, *mexico*, *italian*, *italy*, *spanish*, *german*, *french*, *france*, *english*, *british*, *england*, *russian*, *swiss*, *hawaiian*, *thai*, *brazil*, *brazilian*, *canadian*, *canada*, *australian*, *australia*, *new zealander*, *new zealand*, *korean*, *korea*, *filipino*, *philippines*, *vietnamese*, *vietnam*, *malaysian*, *malaysia*, *singaporean*, *singapore*, *indonesian*, *indonesia*, *thai*, *thailand*, *burmese*, *myanmar*, *cambodian*, *cambodia*, *laotian*, *laos*, *taiwanese*, *taiwan*, *pacific*, *melanesian*, *melanesia*, *polynesian*, *polynesia*, *micronesian*, *micronesia*, *aboriginal*, *aborigine*. We adopt racial classes used in FairFace (*i.e.,* East Asian, Southeast Asian, White, Black, Hispanic, Middle Eastern, Indian), creating attribute-specific descriptions. Specifically, we add *East Asian* for the East Asian class, *Southeast Asian* for the Southeast Asian class, *White* or *Caucasian* for the White class, *Black* or *African* for the Black class, *Latino*, *Latina*, or *Hispanic* for the Hispanic class, *Middle Eastern* or *Arab* for the Middle Eastern class, *Indian* for the Indian class before the person-specific terms (*e.g., person → {East Asian person*, *Southeast Asian person*, *White person*, *Black person*, *Latino person*, *Middle Eastern person*, *Indian person}*).

# D  LIST OF CONCEPTS AND OCCUPATIONS

We show the concepts used to compute MaxSkew@$k$ (in Tab. 1 and 2 in the main paper). The list of the concepts for each category (*i.e.,* adjectives, occupations, and activities) are as follows:

**Adjectives:** *kind*, *successful*, *noncriminal*, *nonviolent*, *moral*, *rich*, *honest*, *smart*, *evil*, *unkind*, *unsuccessful*, *criminal*, *violent*, *immoral*, *poor*, *dishonest*, *dumb*, *clever*, *stupid*, *hardworking*, *lazy*, *nasty*, *trustworthy*, *untrustworthy*, *caring*, *dangerous*, *heroic*, *villainous*, *good*. We refer these concepts from (Berg et al., 2022). Regardig the templates used to create prompts, we use "This is the face of a {} person", "This is a photo of a {} person", "This person is {}", "A {} person", "A {} individual", "This individual is {}", "Photo of a {}", "{}", "This is a {}", "This is the photo of a {}", "A {}", following (Berg et al., 2022). The curly brackets are replaced with each concept.

**Occupations:** *doctor*, *aerospace engineer*, *computer programmer*, *electrical engineer*, *scientist*, *artist*, *author*, *designer*, *musician*, *painter*, *photographer*, *singer*, *writer*, *architect*, *civil engineer*,

*engineer, programmer, software developer, childcare worker, coach, dental assistant, dental hygienist, dentist, clerk, housekeeper, maid, massage therapist, nurse, therapist,psychologist, social assistant, social worker, teacher, professor, CEO, skateboarder, surfer, baseball player, football player, soccer player, tennis player.* For the occupation list, we refer to (Friedrich et al., 2023) with some modifications, such as additional occupations for a more inclusive list. As for the prompt templates, we select ones that are suitable to occupations (*e.g.,* "Photo of a {}"). We use this list for the text-to-image experiments with some modifications. Specifically, we remove the similar occupations (*e.g.,* remove *civil engineer* as there is *engineer*). Additionally, we remove occupations where the text-to-image model (Rombach et al., 2022) can not generate images with humans or it tends to generate multiple individuals.

**Activities:** arranging flowers, playing tennis, playing skateboarding, playing baseball, playing soccer, playing football, playing snowboarding, playing skiing, cleaning, dressmaking, tying tie, smiling, crying, laughing, cooking, making pizza, dancing, drinking beer, drinking wine, eating hotdog, eating cake, using computer, playing game, gardening, singing, petting dog, petting cat, makeup, shopping, playing piano, playing guitar, carrying baby. For activities, we use a subset of the Kinetics dataset (Kay et al., 2017) with some additional activities. For the prompt templates, we use "This is the face of a person who likes {}", " This is a photo of a person who likes {}", "This person likes {}", "A person who likes {}", "Photo of a person who likes {}", "This is a person who likes {}".

## E ADDITIONAL VISUAL EXAMPLES FOR IMAGE GENERATION EXPERIMENTS

We show additional visual examples for Figures 3 and 4 in the main paper in Figures 5 and 6, respectively.

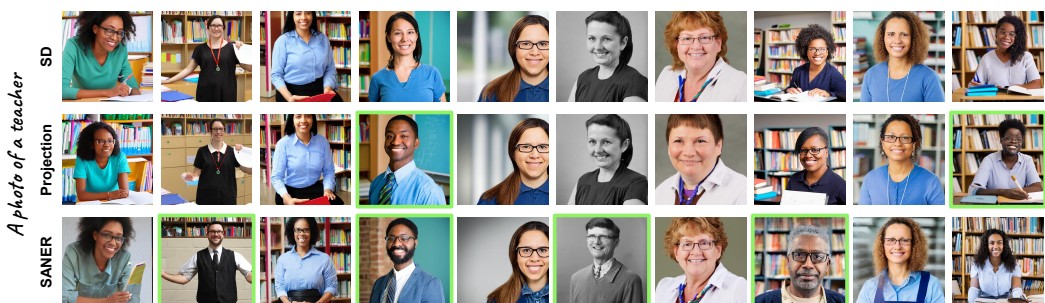

Figure 5: Generated images for the prompt, "A photo of a teacher," by the original Stable Diffusion (SD), projection-based debiased CLIP (Projection), and our debiased CLIP (SANER). We randomly sample 10 images from generated images. Images framed in green denote those of the minority gender in the generated images (*i.e.,* male).

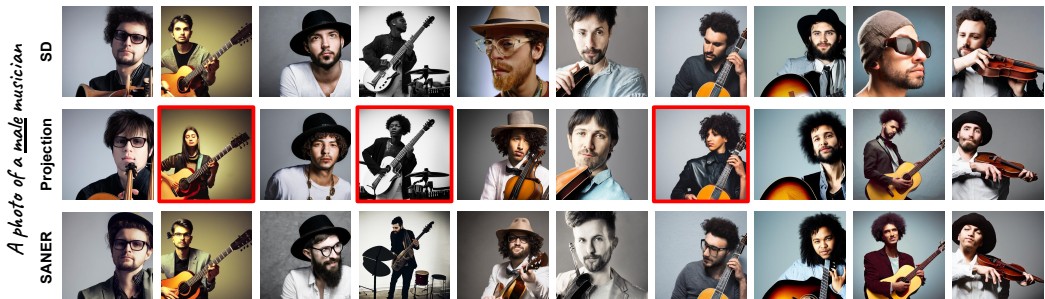

Figure 6: Generated images for the prompt, "A photo of a male musician," by the original Stable Diffusion (SD), projection-based debiased CLIP (Projection), and our debiased CLIP (SANER). We randomly sample 10 images from generated images. Red frame indicates images with incorrect gender (*i.e.,* female).

# F    POTENTIAL EXTENSION

**Additional Attributes.** While we have aimed to make the attribute list as comprehensive as possible based on prior work (Berg et al., 2022; Seth et al., 2023; Chuang et al., 2023), we acknowledge that the choice of attribute groups may be subjective and that certain attributes might be missed. However, in contrast to previous approaches that require extensive attribute labels for images in the dataset, our method is designed to be flexible and extensible, allowing the attribute list to be expanded or adjusted based on specific needs or ethical considerations of the application domain.

**Non-binary gender.** While our experiments followed prior work (Berg et al., 2022; Seth et al., 2023; Chuang et al., 2023) in focusing on binary gender, SANER naturally extends to non-binary gender by defining additional attribute-specific terms and their neutral forms (*e.g.,* "non-binary person" → "person"). The debiasing loss (Eq. 10) can handle any number of attribute groups, making it straightforward to include non-binary gender categories.

**Automated text neutralization.** While our current word-level approach is effective for well-defined social biases, it could be extended to handle more complex cases. As a potential future direction, embedding-based neutralization or context-aware language models could be explored to automate this process. These methods would enable bias identification and neutralization at a semantic level, reducing reliance on predefined attribute word lists and making the debiasing process more robust and adaptable to diverse scenarios.

**Debiasing LLaVA-like models.** SANER can be extended to mitigate societal biases in large vision-language models, such as LLaVA (Liu et al., 2024a), by incorporating a debiasing mechanism for the image encoder. Specifically, this could involve training a debiasing layer to remove attribute information from the visual features, particularly for images without human subjects. We leave this extension as an avenue for future work.

# G    POTENTIAL NEGATIVE IMPACT

Applying SANER to debias CLIP may lead to a potential negative impact where users might overlook remaining biases, assuming the process to be fully effective. While SANER performs better in gender and age bias mitigation, as evidenced by the MaxSkew and statistical parity metrics, this does not ensure that SANER mitigates all possible societal biases, and there could be dimensions of bias not adequately measured by these metrics. It is crucial to acknowledge that SANER, though impactful, is not an all-encompassing solution for removing societal bias. We notice that researchers must exercise due diligence in evaluating the application of SANER to avoid inadvertently introducing unanticipated biases.

