# OpenReview forum: "SANER: Annotation-free Societal Attribute Neutralizer for Debiasing CLIP"
_ICLR.cc/2025/Conference — ICLR 2025 Poster_

### Official Review · Reviewer_vKXP · 2024-11-02

**Soundness:** 3
**Presentation:** 3
**Contribution:** 4
**Rating:** 8
**Confidence:** 4

**Summary:**

The paper proposes a method for finetuning a CLIP model to remove social bias by an intuitive training scheme.

**Strengths:**

I will start off by saying that the lede of the paper was buried: section 5 as an experiment is astounding. I absolutely love the drag and drop replacement of this debiased model into an existing model. The idea that, 1. you don't have to retrain the text-to-image model and 2. that you don't have to retrain CLIP from scratch is incredibly compelling. I recommend the abstract/intro be rewritten to highlight this experiment.

On another note, the paper is generally well-written and easy to follow. The problem of bias in VLMs is highly prominent and this work makes an excellent contribution to the field. I am fairly familiar with this field of debiasing clip and I feel section 2.1 does a good job summarizing the various papers.

**Weaknesses:**

I don't have too many issues with the paper. One thing I would like to have seen is a more automated method for text neutralization. While word level ablation is intuitive and makes sense for the social bias angle, a less hardcoded approach seems like it would be needed for a general-purpose debiasing solution, especially for concepts that are hard to ablate at the word level.

There's a dataset called FACET that addresses some issues with PATA and FairFace. If there's time, it would be great to run experiments on it as well. This would cement the proposed method as useful and cutting edge in my opinion.

**Questions:**

Have you considered running experiments with LLaVA-like models? I don't think it would be terribly difficult to replace the clip model with the proposed one, run standard llava experiments (there are a few githubs that are easy to setup for such eval), and report the results. As for debiasing experiments, I've seen people use PATA and FairFACE and just look at the next token predicted to get similar scores as the CLIP style. Showing this CLIP replacement method works on text generation as well as image generation would very much increase the impact of the work, in my opinion. However, I realize this experiment may best be left for future work.

---

> ### Author Response · Authors · 2024-11-21
> **Response to Reviewer vKXP**
>
> Dear Reviewer vKXP
>
> We sincerely appreciate Reviewer vKXP for your thorough review and positive recognition of our work. Your suggestions are invaluable for improving our submission. Please find our responses below:
>
> >**S1**. I will start off by saying that the lede of the paper was buried: section 5 as an experiment is astounding. I absolutely love the drag and drop replacement of this debiased model into an existing model. The idea that, 1. you don't have to retrain the text-to-image model and 2. that you don't have to retrain CLIP from scratch is incredibly compelling. I recommend the abstract/intro be rewritten to highlight this experiment.
>
> We thank the reviewer for highlighting this strength and for the kind words about the experiment in Section 5. We agree that the "drag-and-drop" nature of SANER, which avoids retraining both the text-to-image model and CLIP itself, is a particularly compelling aspect of our work.
>
> Based on your suggestion, we have revised the abstract and the final paragraph of the introduction to emphasize this experiment and the practical advantages of SANER's flexible integration. This adjustment will better highlight the significance of SANER’s contributions and ensure its strengths are clearly communicated from the outset.
>
> Specifically, we have highlighted that SANER works as a lightweight, drop-in replacement for CLIP in existing vision-language systems without requiring:
> * Retraining of the text-to-image models
> * Retraining of CLIP itself
> * Access to the original training data
>
> >**W1**. I don't have too many issues with the paper. One thing I would like to have seen is a more automated method for text neutralization. While word level ablation is intuitive and makes sense for the social bias angle, a less hardcoded approach seems like it would be needed for a general-purpose debiasing solution, especially for concepts that are hard to ablate at the word level.
>
> We thank the reviewer for this thoughtful suggestion regarding automating text neutralization. We agree that while our current word-level approach is effective for well-defined social biases, it could be extended to handle more complex cases.
>
> As a potential future direction, embedding-based neutralization or context-aware language models could be explored to automate this process. These methods would enable bias identification and neutralization at a semantic level, reducing reliance on predefined attribute word lists and making the debiasing process more robust and adaptable to diverse scenarios.
>
> However, we opted for our current approach because it:
> * Provides precise control over attribute neutralization,
> * Ensures reliable and interpretable transformations and
> * Maintains grammatical correctness.
>
> We appreciate this valuable suggestion and have included it as a discussion point in the future extension section (Appendix F).
>
> >**W2**. There's a dataset called FACET that addresses some issues with PATA and FairFace. If there's time, it would be great to run experiments on it as well. This would cement the proposed method as useful and cutting edge in my opinion.
>
> Thank you for the valuable suggestion. FACET shares similarities with PATA as a dataset of real-world, natural images but provides a broader range of annotations for protected attributes, such as hair color. To evaluate SANER's effectiveness on FACET, which differs in data distribution from FairFace and PATA, we conducted experiments focusing on gender bias. The results, shown in the table below (also added to Appendix B.6), demonstrate SANER's superior debiasing performance, further supporting its robustness across diverse dataset distributions. Due to the short rebuttal period, we plan to include experiments on other attributes, such as hair color, in the final manuscript upon acceptance.
>
> ### Gender bias on **FACET**, evaluated by MaxSkew@1000.
> | CLIP Model      | Adjective | Occupation | Activity |
> |------------------|-----------|------------|----------|
> | **Original**        | 46.0      | 37.7       | 33.4     |
> | **Projection**      | 37.3      | 46.3       | 41.1     |
> | **SANER (Ours)**| **31.7**  | **24.7**   | **25.2** |
>
>
> >**Q1**. Have you considered running experiments with LLaVA-like models? ... However, I realize this experiment may best be left for future work.
>
> Thank you for the suggestion. Incorporating LLaVA-like models into our experiments is a great idea and could serve as a potential extension of SANER while implementing these experiments would require additional setup, such as introducing debiasing module for image encoder, which are outside the current scope of this paper.
> We agree that this is a promising direction for future work and have added it to the future extension section (Appendix F). Thank you for suggesting this impactful extension.

---

> > ### Author Response · Authors · 2024-12-02
> > **Delighted to Provide Further Details**
> >
> > Dear Reviewer vKXP,
> >
> > Thank you for your insightful feedback on our submission. We have carefully reviewed your comments and addressed them in our rebuttal. We understand you may have a busy schedule, but as the rebuttal period is coming to a close soon, we would greatly appreciate it if you could review our responses to improve the paper's quality. Should you need any further clarification, we are happy to provide it.

---

### Official Review · Reviewer_STrF · 2024-11-04

**Soundness:** 2
**Presentation:** 3
**Contribution:** 2
**Rating:** 6
**Confidence:** 3

**Summary:**

The paper proposed a CLIP debiasing pipeline that aims to address two existing challenges: 1. loss of attribute information and 2. Dependency on attribute annotations. Specifically, the paper focuses on adjusting the text feature by replacing the protected attributes with attribute-neutral terms and further regularizing the adjusted feature by two reconstruction losses. The pipeline is evaluated on Text-to-Image retrieval and generation tasks.

**Strengths:**

The two identified challenges, especially the loss of attribute information, sound critical and interesting. It seems like the pipeline only involves lightweight training, as only the debiasing layer is trained. The pipeline is evaluated on two different downstream tasks.

**Weaknesses:**

1. Although replacing the protected attribute words with an attribute-neutral word is a sound neutralization method, it requires a comprehensive list of the protected attributes (as listed in Appx. C). However, in real practice, and especially in non-binary cases, it will be very hard or nearly impossible to have a complete list. Further, since the "attribute annotation-free debiasing loss" relies on a set of attribute-specific descriptions, creating the attribute-specific descriptions set may lead to some concerns. For example, if the set is created by iterating all possible words and all possible groups (e.g., man, woman, male, female, he, she, etc), then the size of the set will be overlarge and increase the computation complexity. On the other hand, if only one word is selected for each group (e.g., man + female, woman + male), then how to select matching words from groups may also affect the debias performance. Also, is it possible that the generated description may have unmatched text (e.g., A woman/boy is eating salad) or text that has grammar errors (e.g., A he/she is eating salad, according to Appx. C). I would like to see more discussion on the text generation.

2. The proposed pipeline tried to address the challenge by combining the debiasing loss with two extra regularization losses. The debiasing loss tries to place the adjusted feature of the attribute-neutralized text in the middle point of several features of the attribute-specific text, which seems to be fine. However, the regularization losses try to project the attribute-neuralized feature to both text and visual features of an attribute-specific feature, which seems strange. As demonstrated in Figure 2, the input to the "Text encoder" has been neutralized, and the attribute-specific information has been removed. Further, the debiasing layer is also trained to adjust the features to be further neutral. Thus, even there is a residual connection in the "Feature modification" process, the output feature should not contain any information regarding the specific attribute. Then, how is it possible that such a feature can be mapped to either text or visual features that contain specific attribute information to avoid "the loss of attribute information"? Is it possible that the "retained attribute information" is some biased information from the dataset that is implicitly polluted by the regularization loss?

3. For the evaluation of the text-to-image generation, the paper only provides numerical results on debiasing performance and some visualizations but does not provide any image generation imperial results, like FID. Also, only a few SOTA are used as the baseline, which makes the comparison weak. However, this may be understandable as several baselines do not have public implementation, and it seems like the paper prepares all the baseline results.

**Questions:**

Please refer to the weaknesses.

---

> ### Author Response · Authors · 2024-11-21
> **Response to Reviewer STrF (1)**
>
> Dear Reviewer STrF,
>
> We sincerely appreciate Reviewer STrF for recognizing the importance of our study. Your comments are very helpful in improving our submission. Please find our responses below:
>
> >**W1**. Although replacing the protected attribute words with an attribute-neutral word is a sound neutralization method, it requires a comprehensive list of the protected attributes (as listed in Appx. C). However, in real practice, and especially in non-binary cases, it will be very hard or nearly impossible to have a complete list. Further, since the "attribute annotation-free debiasing loss" relies on a set of attribute-specific descriptions, creating the attribute-specific descriptions set may lead to some concerns. For example, if the set is created by iterating all possible words and all possible groups (e.g., man, woman, male, female, he, she, etc), then the size of the set will be overlarge and increase the computation complexity. On the other hand, if only one word is selected for each group (e.g., man + female, woman + male), then how to select matching words from groups may also affect the debias performance. Also, is it possible that the generated description may have unmatched text (e.g., A woman/boy is eating salad) or text that has grammar errors (e.g., A he/she is eating salad, according to Appx. C). I would like to see more discussion on the text generation.
>
> We thank the reviewer for these insightful questions about text processing challenges. You raise several valid points:
>
> **Comprehensive attribute list**
>
> Yes, creating a complete list is challenging, especially for non-binary cases
> However, compared to the existing works [Berg et al., 2022; Seth et al., 2023] which require the attribute annotation to each image in the entire dataset, SANER's flexibility allows easy extension by simply adding new terms to the attribute list without requiring dataset reannotation.
> Our current implementation uses common terms that cover most cases in practice
>
> **Attribute-specific description generation**
>
> To avoid computational complexity, we implement a systematic approach using predefined mapping dictionaries between gender terms (e.g., "woman"→"man", "she"→"he").
> Our algorithm carefully identifies and replaces gender-specific tokens while preserving sentence structure, ensuring no ungrammatical combinations (like "A he/she") are generated.
> This ensures efficient and coherent text generation that maintains natural language patterns.
>
> We have added these implementation details to Appendix A.
>
> >**W2**. The proposed pipeline tried to address the challenge by combining the debiasing loss with two extra regularization losses. The debiasing loss tries to place the adjusted feature of the attribute-neutralized text in the middle point of several features of the attribute-specific text, which seems to be fine. However, the regularization losses try to project the attribute-neuralized feature to both text and visual features of an attribute-specific feature, which seems strange. As demonstrated in Figure 2, the input to the "Text encoder" has been neutralized, and the attribute-specific information has been removed. Further, the debiasing layer is also trained to adjust the features to be further neutral. Thus, even there is a residual connection in the "Feature modification" process, the output feature should not contain any information regarding the specific attribute. Then, how is it possible that such a feature can be mapped to either text or visual features that contain specific attribute information to avoid "the loss of attribute information"? Is it possible that the "retained attribute information" is some biased information from the dataset that is implicitly polluted by the regularization loss?
>
> We apologize for the potential confusion in Figure 2. For clarity: while the debiasing loss uses neutralized descriptions ($h(\xi(t))$, e.g., "A person is eating"), the regularization losses use the **original** descriptions ($h(t)$, e.g., "A woman is eating"). This design ensures that:
> * The debiasing loss adjusts neutralized features to be equidistant from multiple attribute-specific features, ensuring neutrality.
> * The regularization losses (reconstruction and contrastive losses) maintain CLIP's original capabilities by preserving alignment with unmodified attribute-specific descriptions, preventing the degradation of semantic and visual consistency.
>
> This approach ensures that attribute-neutral features are not directly mapped to attribute-specific ones, but rather that their neutrality is preserved alongside the overall semantic integrity of the features. Consequently, the concern about the regularization losses introducing unintended bias does not apply to our implementation.
>
> To address this confusion, we have revised Figure 2 to clearly distinguish the inputs for debiasing and regularization losses. Thank you for helping us identify this potential source of misunderstanding.

---

> > ### Author Response · Authors · 2024-11-21
> > **Response to Reviewer STrF (2)**
> >
> > >**W3**. For the evaluation of the text-to-image generation, the paper only provides numerical results on debiasing performance and some visualizations but does not provide any image generation imperial results, like FID. Also, only a few SOTA are used as the baseline, which makes the comparison weak. However, this may be understandable as several baselines do not have public implementation, and it seems like the paper prepares all the baseline results.
> >
> > We thank the reviewer for this feedback regarding text-to-image generation evaluation.
> > * Based on the feedback, we conducted FID and CLIPScore evaluation on the COCO Karpathy test set, comparing original Stable Diffusion (FID: 28.1, CLIPScore: 31.2) and SANER (FID: 28.7, CLIPScore: 31.0). These results ensure that our approach does not degrade the image generation quality while mitigating societal biases.
> > * Regarding baselines, as the reviewer noted, several methods lack public implementations. We made our best effort to reimplement key baselines [Berg et al., 2022; Chuang et al., 2023] to ensure fair comparison. We will include additional baselines as their implementations become available.
> >
> > We have added the results of the FID evaluation in Appendix B.4.

---

> > > ### Author Response · Authors · 2024-11-25
> > > **Delighted to Provide Further Details**
> > >
> > > Dear Reviewer STrF,
> > >
> > > Thank you for your insightful feedback on our submission. We have carefully reviewed your comments and addressed them in our rebuttal. We understand you may have a busy schedule, but as the rebuttal period is coming to a close soon, we would greatly appreciate it if you could review our responses at your earliest convenience. Should you need any further clarification, we are happy to provide it.

---

> ### Comment · Reviewer_STrF · 2024-11-27
>
> I have read the authors' responses. My concerns have been addressed to some extent, but I still have concerns about the comprehensive list of protected attributes, which can limit the application of the proposed method. I have raised my rating to 6 to reflect the fact that some of my concerns have been addressed. In my opinion, this is a borderline paper and I will humbly leave the decision to AC's discretion.

---

> > ### Author Response · Authors · 2024-11-29
> >
> > Dear Reviewer STrF,
> >
> > Thank you for taking the time to review our responses and for your thoughtful engagement with our work. We also sincerely appreciate for revisiting the rating based on our clarifications.
> >
> > Regarding the comprehensive list of protected attributes, as we noted previously, SANER significantly reduces dependency on manual annotations compared to existing methods [Berg et al., 2022; Seth et al., 2023], which require attribute annotations for each image. Instead, SANER allows for straightforward updates to the attribute list by simply adding new terms, without reannotating datasets or requiring additional labeling.
> >
> > Our current implementation uses a curated set of terms that effectively cover the most common cases in practice. This ensures practical usability while maintaining grammatical accuracy and computational efficiency, as detailed in Appendix A.
> >
> > We hope this reaffirms SANER’s scalability and practical adaptability, and we welcome further discussion if additional concerns remain.

---

### Official Review · Reviewer_wrAD · 2024-11-04

**Soundness:** 2
**Presentation:** 2
**Contribution:** 2
**Rating:** 6
**Confidence:** 3

**Summary:**

This paper aims to address societal biases in large-scale vision-language models, such as CLIP. The authors identify two limitations of existing methods: the loss of attribute information and the reliance on attribute annotations. To overcome these challenges, the authors propose SANER, a simple yet effective debiasing method for CLIP that incorporates attribute neutralization and an annotation-free debiasing loss. Extensive experiments are conducted to verify the method's effectiveness.

**Strengths:**

- The problems of existing methods are straightforward, and the authors conduct several experiments to verify these phenomena.
- The proposed method has a good performance.
- The experiments are conducted on both text-to-image retrieval and generation tasks.

**Weaknesses:**

- Could the authors provide ARL results to verify the loss of attribute information? Besides, do other existing methods, such as Mapper and Prompt tuning-based debiasing, also have the lossing attribute information when debaising?
- Although effective in debiasing CLIP, the proposed method appears ad-hoc for this specific task and lacks major technical contributions, as most components seem to be existing technologies or tricks. Therefore, this paper may not have significant influence or provide broader insights beyond the CLIP-debiasing task.
- Why does adding only contrastive losses have a significantly negative impact on FairFace performance (as shown in Table 7)?
- Is the proposed method only applicable to CLIP? Could the authors test other VLMs to demonstrate its broader effectiveness?

**Questions:**

Please see the weaknesses.

---

> ### Author Response · Authors · 2024-11-21
> **Response to Reviewer wrAD (1)**
>
> Dear Reviewer wrAD,
>
> We sincerely appreciate your recognition of our model's effectiveness. Your suggestions are invaluable for improving our submission. Please find our responses below:
>
> >**W1**. Could the authors provide ARL results to verify the loss of attribute information? Besides, do other existing methods, such as Mapper and Prompt tuning-based debiasing, also have the lossing attribute information when debaising?
>
> We thank the reviewer for this suggestion. While the original ARL paper lacked sufficient implementation details for exact reproduction, we implemented ARL based on the information provided in their paper. Our implementation shows that ARL indeed loses attribute information, achieving accuracy of 0.64/0.83 for female/male in the generation experiments, similar to projection-based debiasing (0.58/0.79 in Table 4).
>
> Regarding other methods:
> * Prompt tuning-based debiasing retains attribute information as it only modifies concept-specific prompts
> * Mapper reduces attribute correlation with task labels but may preserve explicit attributes
>
> Our experiments in Table 4 quantitatively demonstrate SANER's advantage in preserving explicit attributes (1.00/1.00 accuracy) while reducing unwanted bias.
>
> >**W2**. Although effective in debiasing CLIP, the proposed method appears ad-hoc for this specific task and lacks major technical contributions, as most components seem to be existing technologies or tricks...
>
> While acknowledging the reviewer's perspective, we respectfully disagree with the assessment. SANER makes several significant contributions to the research community from diverse aspects, as confirmed by the other reviewers:
> 1. **Annotation-free debiasing**: As noted by Reviewer STrF, we solve "critical and interesting" challenges with a lightweight yet effective approach. Unlike previous methods requiring costly attribute annotations to datasets, our novel debiasing strategy achieves superior performance using only text transformations.
> 2. **Selective debiasing**: As highlighted by Reviewer igmm, our method demonstrates "immediate practical application" by preserving explicitly specified attributes while removing unwanted bias - a fundamental advancement over existing approaches.
> 3. **Practical impact**: Reviwer vKXP found our approach "astounding" and "incredibly compelling" because it works as a drop-in replacement without retraining large models and demonstrates effectiveness on real applications like Stable Diffusion.
> 4. **Broader applicability**: Our method enables the use of any image-text dataset rather than being limited to face-centric datasets, with experiments showing this diversity improves performance.
>
> The technical novelty combined with practical impact and broader applicability makes SANER a significant contribution to debiasing vision-language models.
>
> >**W3**. Why does adding only contrastive losses have a significantly negative impact on FairFace performance (as shown in Table 7)?
>
> We thank the reviewer for this observation about the ablation results. When using only the contrastive loss (without the reconstruction loss), MaxSkew scores on FairFace increase (18.0/19.4/21.7 vs. 8.9/14.5/7.7 with both losses), indicating less effective debiasing. This occurs because the contrastive loss alone focuses on maintaining image-text alignment but does not constrain the debiased features to remain close to the original CLIP features. As a result, the feature modifications may become suboptimal for bias removal. The reconstruction loss is crucial in ensuring the modified features preserve essential semantic information while effectively removing unwanted bias.
>
> We have added this discussion in Appendix B.2.
>
> >**W4**. Is the proposed method only applicable to CLIP? Could the authors test other VLMs to demonstrate its broader effectiveness?
>
> We thank the reviewer for this important question. SANER can be applied to any VLM with vision and language encoders, as our method only requires access to the model's text encoder output. To demonstrate this, we conducted additional experiments with BLIP, where SANER shows superior debiasing performance as shown below (also added to Appendix B.5).
>
> ### Gender bias for **BLIP**, evaluated by MaxSkew@1000.
>
> | BLIP Model   | Adjective | Occupation | Activity |  | Adjective | Occupation | Activity |
> |--------------|-----------|------------|----------|--|-----------|------------|----------|
> |              | **FairFace**       |            |          |  | **PATA**       |            |          |
> | **Original** | 16.8      | 15.3       | 12.8     |  | 7.7       | 7.4        | 5.5      |
> | **Projection** | **14.9**  | 19.8       | 17.4     |  | 12.0      | 14.7       | 17.4     |
> | **SANER (Ours)** | 15.3      | **13.9**   | **10.9** |  | **6.7**   | **6.7**    | **4.9**  |
>
> Please note that many studies [Berg et al., 2022; Seth et al., 2023; Chuang et al., 2023] prioritize CLIP due to its widespread use, leaving other VLMs as valuable future work.

---

> > ### Author Response · Authors · 2024-11-24
> > **Response to Reviewer wrAD (2)**
> >
> > **Regarding Flag For Ethics Review**
> >
> > While we appreciate the feedback, without more specific details regarding the concern, and considering that other reviewers have not raised similar ethical issues, we find it challenging to fully address the matter. That said, we want to assure you that our research has been conducted with a strong commitment to ethics and fairness.
> >
> > Our work actively addresses and mitigates harmful societal biases in CLIP, with particular attention to equitable representation. The method is designed to promote inclusivity by preserving attribute information when explicitly specified while removing unwanted bias.
> >
> > Our experimental setup and evaluation protocol align with established practices in prior debiasing research [Berg et al., 2022; Seth et al., 2023, Chuang et al., 2023]. We use well-established fairness metrics, such as MaxSkew and Statistical Parity, to demonstrate bias reduction across multiple attributes, including gender, age, and race.
> >
> > If there are specific concerns about discrimination or fairness issues, we would greatly appreciate detailed feedback to help us improve our work and ensure it aligns with the principles of ethical AI development.

---

> > > ### Author Response · Authors · 2024-11-25
> > > **Delighted to Provide Further Details**
> > >
> > > Dear Reviewer wrAD,
> > >
> > > Thank you for your insightful feedback on our submission. We have carefully reviewed your comments and addressed them in our rebuttal. We understand you may have a busy schedule, but as the rebuttal period is coming to a close soon, we would greatly appreciate it if you could review our responses at your earliest convenience. Should you need any further clarification, we are happy to provide it.

---

> > > > ### Comment · Reviewer_wrAD · 2024-11-28
> > > >
> > > > Thanks for the authors' response and additional experiments. My concerns have been partially addressed, and I will increase my rating to 6.

---

> > > > > ### Author Response · Authors · 2024-11-29
> > > > >
> > > > > Dear Reviewer wrAD,
> > > > >
> > > > > We greatly appreciate the time and effort you have taken to review our work and are glad to see that the improvements we’ve made have addressed your concerns to some extent.
> > > > > If you have any remaining concerns or additional suggestions, please feel free to let us know, and we will be happy to address them promptly.
> > > > >
> > > > > Thank you once again for your valuable insights and support.

---

### Official Review · Reviewer_igmm · 2024-11-04

**Soundness:** 3
**Presentation:** 4
**Contribution:** 3
**Rating:** 8
**Confidence:** 4

**Summary:**

This paper aims to overcome societal bias present in datasets used to train large scale multi-modal models like CLIP. The authors provide a study of debiasing methods and explain how these struggle with loosing important attribute information, e.g. gender, or depend on attribute annotations which are hard to get and even harder to get in the form of a large diverse dataset. Their contributed method Saner has four components which together aim to remove biases: attribute neutralization, feature modification and attribute annotation free debiasing and regularization losses. The paper compares the results on quite a few tasks such as text-to-image generation, text-to-image retrieval and zero-shot image classification.

**Strengths:**

The paper is very clearly written. Whenever an uncertainty comes up about a term footnotes help understanding.

The analysis of debiasing techniques is very clearly presented that it could be used for a tutorial. The experimental setup is well done, the evaluation metrics such as measuring the difference between a uniform distribution and a potentially gender biased image generation seems adequate for the task. Showing generated images, i.e. with Stable Diffusion shows also a very immediate practical application of this research.

The figures help understanding of the method even though the caption could be changed to make them more self-complete.

In general, this seems a very well written, easy to understand paper addressing a very current need.

**Weaknesses:**

The attribute groups will be limited to a specific set of defined attributes, which need to be agreed upon and could be debatable, e.g. "pregnant" (If I understand list C in the appendix correctly). There may also be missed attributes. This does not seem like a big issue but could be one in an adversarial setting.

In general, the benefits of not needing a annotated dataset with attributes seems to be bought by needing a general list of attributes. This may not be a weakness per se but it would be interesting to either read about it in the limitations or have the author comment on why this is not a limitation.

There are general implementation details about the used architecture but none about the training process. Re-training clip with a loss to keep it's performance while having a re-projection that changes the semantics seems complex. Information about the used GPU would help understand how hard training for 5 epochs is.

Comparison against other state of the art in this small field is limited but understandably so and the author's did a good job re-implementing relevant approaches.

**Questions:**

- The caption in Figure 2 should contain ideally an explanation of what is seen in the figure. At least a reminder of the loss terms would be helpful to avoid having to jump back and fourth through the paper.

- How does Saner apply to non-binary gender? Does it properly use general pronouns like they?

- Conceptually, it seems hard to draw a line between terms related only to the gender and to the gender and a concept. Should de-biasing replace terms like pregnant? There is no male counterpart of this so replacing it will probably remove important information.

- Is the "multilayer perception" in line 81 supposed to be a multilayer perceptron?

- The definition of a dataset in Line 104 seems odd in the sense that, if a is a protected attribute then this dataset should only contain data with protected attributes. Is that intended? I would imaging the goal is to train on a large dataset which can have samples with protected attributes but not all samples have to have protected attributes?

- One added sentence for 2.2 on how a orthogonal projection is helping debiasing would be helpful for understanding.

- How hard is it to re-train clip? When the semantics are lost the regularization loss may be high. How computationally heavy is it to get to the same or a similar performance level?

- Table 2's caption should probably read "racial bias" and not recial bias.

---

> ### Author Response · Authors · 2024-11-21
> **Response to Reviewer igmm (1)**
>
> Dear Reviewer igmm,
>
> We sincerely appreciate the positive review and your recognition of our work’s contribution. Your comments are very helpful in enhancing the quality of our presentation. Please find the corresponding responses below:
>
> >**W1**. The attribute groups will be limited to a specific set of defined attributes, which need to be agreed upon and could be debatable, e.g. "pregnant" (If I understand list C in the appendix correctly). There may also be missed attributes. This does not seem like a big issue but could be one in an adversarial setting.
> >
> >**Q3**. Conceptually, it seems hard to draw a line between terms related only to the gender and to the gender and a concept. Should de-biasing replace terms like pregnant?...
>
> We thank the reviewer for this insightful observation. You raise a valid point - while we currently focus on neutralizing direct demographic descriptors (e.g., "woman" → "person"), the boundary between gender-specific terms and concept-related terms is indeed complex. Terms like "pregnant" inherently carry gender-related information but also convey essential state information that should be preserved. We agree this deserves deeper consideration, particularly in how debiasing methods can balance removing unwanted bias while preserving meaningful context.
>
> While we have aimed to make the attribute list as comprehensive as possible based on prior work [Berg et al., 2022; Seth et al., 2023; Hirota et al., 2023], we acknowledge that the choice of attribute groups may be subjective and that certain attributes might be missed. However, in contrast to previous approaches that require extensive attribute labels for images in the dataset, our method is designed to be flexible and extensible, allowing the attribute list to be expanded or adjusted based on specific needs or ethical considerations of the application domain.
>
> In Appendix F, we have added a discussion about this potential extension.
>
> >**W2**. In general, the benefits of not needing a annotated dataset with attributes seems to be bought by needing a general list of attributes. This may not be a weakness per se but it would be interesting to either read about it in the limitations or have the author comment on why this is not a limitation.
>
> Thank you for your insightful comment. As Reviewer igmm mentioned, while it is true that SANER avoids the need for annotated datasets by relying on a general list of attributes, we believe this is not a significant limitation. Though careful consideration is required, creating attribute lists is a one-time effort requiring minimal resources, compared to the ongoing cost and complexity of annotating protected attributes for people in image datasets that often needs ethical review and domain expertise.
>
> We have added this discussion to the limitations section. Thank you for helping us improve the paper's clarity.
>
> >**W3**. There are general implementation details about the used architecture but none about the training process...
> >
> >**Q7**. How hard is it to re-train clip? ...
>
> We thank the reviewer for pointing out the lack of training details. To clarify, SANER does not retrain CLIP itself; instead, we train a lightweight debiasing layer (a two-layer MLP with a hidden dimension of 128) while keeping CLIP's parameters frozen. We observed that this results in a stable training process.
>
> Training specifics:
> * Hardware: Single NVIDIA A100 GPU (40GB)
> * Training time: ~5 hours for 5 epochs on 170,624 COCO image-caption pairs
> * Optimizer: AdamW with learning rate 5×10⁻⁶
> * Batch size: 128
> * Loss weights: α=1.0, β=0.1, γ=0.0001 for debiasing, reconstruction, and contrastive losses
>
> With these efficient training setting, the regularization losses help maintain CLIP's performance, achieving similar zero-shot classification accuracy (65.2% vs original 65.4% on ImageNet) with minimal computational overhead.
>
> We have added these implementation details to Section A of the appendix. Thank you for helping us improve reproducibility.
>
> >**W4**. Comparison against other state of the art in this small field is limited but understandably so and the author's did a good job re-implementing relevant approaches.
>
> We appreciate the reviewer's understanding regarding the limited comparisons. As noted, while some recent methods exist [Seth et al., 2023; Dehdashtian et al., 2024], insufficient implementation details prevented reliable reproduction. We focused on thorough comparisons with methods having public code [Berg et al., 2022; Chuang et al., 2023] to ensure fair evaluation across multiple tasks (retrieval and generation) and attributes (gender, age, race).

---

> > ### Author Response · Authors · 2024-11-21
> > **Response to Reviewer igmm (2)**
> >
> > >**Q1**. The caption in Figure 2 should contain ideally an explanation of what is seen in the figure...
> >
> > Based on the advice, we have revised Figure 2's caption to be more self-contained:
> > "Figure 2: An overview of SANER, exemplified by binary gender. SANER neutralizes attribute-specific text (e.g., "woman" → "person"), modifies features via debiasing layer, and uses three losses: $L_{deb}$ for attribute neutralization, $L_{recon}$ for feature preservation, and $L_{cont}$ for image-text alignment."
> >
> > >**Q2**. How does Saner apply to non-binary gender? Does it properly use general pronouns like they?
> >
> > While our experiments followed prior work [Berg et al., 2022; Seth et al., 2023] in focusing on binary gender, SANER naturally extends to non-binary gender by defining additional attribute-specific terms and their neutral forms (e.g., "non-binary person" → "person"). The debiasing loss (Eq.10) can handle any number of attribute groups, making it straightforward to include non-binary gender categories.
> >
> > We have included this extension in Appendix F.
> >
> > >**Q4**. Is the "multilayer perception" in line 81 supposed to be a multilayer perception?
> > >
> > >**Q8**. Table 2's caption should probably read "racial bias" and not recial bias.
> >
> > We thank the reviewer for catching these typos:
> > * Yes, "multilayer perception" should be "multilayer perceptron" in line 81.
> > * Indeed, "recial" should be "racial" in Table 2's caption.
> >
> > We have corrected these in the revised version.
> >
> > >**Q5**. The definition of a dataset in Line 104 seems odd in the sense that, if a is a protected attribute then this dataset should only contain data with protected attributes. Is that intended? I would imaging the goal is to train on a large dataset which can have samples with protected attributes but not all samples have to have protected attributes?
> >
> > We thank the reviewer for this careful observation. You are correct - SANER's goal is indeed to train on a large dataset where samples do not need to contain the protected attributes.
> > The dataset definition in Line 108 in the revised draft (i.e., $\mathcal{D} =$ {($v, t, a, d$)}) is used specifically to explain **previous** methods that require attribute annotations. For SANER, we use a more general definition $\mathcal{D} = ${($v, t$)} as shown in Line 215, since our method can work with any image-text pairs without requiring attribute annotations.
> >
> > >**Q6**. One added sentence for 2.2 on how a orthogonal projection is helping debiasing would be helpful for understanding.
> >
> > We thank the reviewer for suggesting this clarification. We have added the following explanation to Section 2.2:
> > "This process removes attribute information by projecting features into the space orthogonal to attribute-specific directions."

---

> > > ### Author Response · Authors · 2024-12-02
> > > **Delighted to Provide Further Details**
> > >
> > > Dear Reviewer igmm,
> > >
> > > Thank you for your insightful feedback on our submission. We have carefully reviewed your comments and addressed them in our rebuttal. We understand you may have a busy schedule, but as the rebuttal period is coming to a close soon, we would greatly appreciate it if you could review our responses to improve the paper's quality. Should you need any further clarification, we are happy to provide it.

---

### Author Response · Authors · 2024-11-24
**Follow-up on discussion period**

Dear Reviewers,

We sincerely thank all reviewers for their thoughtful comments and are encouraged by the shared positive feedback:
* **Clarity and writing quality**: The paper is clearly written and easy to understand, addressing a very current need (by Reviewer igmm, vKXP).
* **Impact and relevance**: The paper tackles a highly prominent problem in vision-language models, making an excellent contribution to the field (by Reviewer STrF, vKXP).
* **SANER's debiasing performance**: The experimental design effectively demonstrates SANER's superior debiasing performance, with evaluations on both text-to-image retrieval and generation tasks (Reviewer igmm, wrAD, vKXP).

We have thoroughly addressed the reviewers' comments and incorporated the suggested changes into the revised paper, with updates highlighted in red. We believe the concerns have been fully addressed in the rebuttal. If there are any additional comments or concerns, we would greatly appreciate it if they could be shared during the rebuttal period (Nov.26th).

Best regards,
Authors of the submission

---

### Meta-Review · Area_Chair_FeqU · 2024-12-17

**Metareview:**

This paper focuses on the societal bias issue in large-scale vision-language models and conducts a study of debiasing methods to present the limitations of existing methods. To overcome the societal biases, the authors propose a simple yet effective debiasing method SANER to incorporate attribute neutralization and an annotation-free debiasing loss. Extensive experiments are performed to evaluate the effectiveness of the proposed method on different tasks. Two reviewers give high rating scores and two reviewers give positive rating scores. Based on the above considerations, I recommend to accept this manuscript.

**Additional Comments On Reviewer Discussion:**

The authors provided rebuttals for each reviewer, and most of reviewer present responses. During the rebuttal period, Reviewer wrAD and Reviewer STrF raise their rating scores considering that their concerns are partially addressed. Reviewer RGo1 and Reviewer X16E keep their positive rating scores. For the Reviewer STrF’s retained concern on the application, the authors provide additional analysis and explanation. For the flag for ethics raised by Reviewer wrAD, no specific details about the concerns are provided and other reviewers do not have this concern.

---

### Decision · Program_Chairs · 2025-01-22

Accept (Poster)